# Exogenous Sodium Nitroprusside Mitigates Salt Stress in Lentil (*Lens culinaris* Medik.) by Affecting the Growth, Yield, and Biochemical Properties

**DOI:** 10.3390/molecules26092576

**Published:** 2021-04-28

**Authors:** Tauqeer Ahmad Yasir, Ayesha Khan, Milan Skalicky, Allah Wasaya, Muhammad Ishaq Asif Rehmani, Naeem Sarwar, Khuram Mubeen, Mudassir Aziz, Mohamed M. Hassan, Fahmy A. S. Hassan, Muhammad Aamir Iqbal, Marian Brestic, Mohammad Sohidul Islam, Subhan Danish, Ayman EL Sabagh

**Affiliations:** 1College of Agriculture, Bahauddin Zakariya University, Bahadur Sub-Campus Layyah, Layyah 31200, Pakistan; tayasir@yahoo.com (T.A.Y.); ayeshakhan3630@gmail.com (A.K.); 2Department of Botany and Plant Physiology, Faculty of Agrobiology, Food, and Natural Resources, Czech University of Life Sciences Prague, Kamycka 129, 165 00 Prague, Czech Republic; skalicky@af.czu.cz; 3Department of Agronomy, Ghazi University, Dera Ghazi Khan 32200, Pakistan; mrehmani@gudgk.edu.pk; 4Department of Agronomy, Bahauddin Zakariya University, Multan 60000, Pakistan; naeemsarwar@bzu.edu.pk; 5Department of Agronomy, MNS University of Agriculture, Multan 60000, Pakistan; khurram.mubeen@mnsuam.edu.pk (K.M.); mudassiraziz@hotmail.com (M.A.); 6Department of Biology, College of Science, Taif University, P.O. Box 11099, Taif 21944, Saudi Arabia; khyate_99@yahoo.com (M.M.H.); fahmy_hssn@yahoo.com (F.A.S.H.); 7Department of Agronomy, Faculty of Agriculture, University of Poonch Rawalakot, Rawalakot 12350, Pakistan; aamir1801@yahoo.com; 8Department of Plant Physiology, Slovak University of Agriculture, Nitra, Tr. A. Hlinku 2, 949 01 Nitra, Slovakia; marian.brestic@uniag.sk; 9Depatment of Agronomy, Hajee Mohammad Danesh Science and Technology University, Dinajpur 5200, Bangladesh; shahid_sohana@yahoo.com; 10Departments of Soil Science, Faculty of Agricultural Sciences and Technology, Bahauddin Zakariya University, Multan, Punjab 60800, Pakistan; sd96850@gmail.com; 11Department of Agronomy, Faculty of Agriculture, Kafrelsheikh University, Kafrelsheikh 33156, Egypt

**Keywords:** antioxidant enzymes, abiotic stress tolerance, mitigation, legumes, reactive oxygen species

## Abstract

Soil salinity disrupts the physiological and biochemical processes of crop plants and ultimately leads to compromising future food security. Sodium nitroprusside (SNP), a contributor to nitric oxide (NO), holds the potential to alleviate abiotic stress effects and boost tolerance in plants, whereas less information is available on its role in salt-stressed lentils. We examined the effect of exogenously applied SNP on salt-stressed lentil plants by monitoring plant growth and yield-related attributes, biochemistry of enzymes (superoxide dismutase (SOD), catalase (CAT), and peroxidase (POD)) amassing of leaf malondialdehyde (MDA) and hydrogen peroxide (H_2_O_2_). Salinity stress was induced by NaCl application at concentrations of 50 mM (moderate salinity) and 100 mM (severe salinity), while it was alleviated by SNP application at concentrations of 50 µM and 100 µM. Salinity stress severely inhibited the length of roots and shoots, the relative water content, and the chlorophyll content of the leaves, the number of branches, pods, seeds, seed yield, and biomass per plant. In addition, MDA, H_2_O_2_ as well as SOD, CAT, and POD activities were increased with increasing salinity levels. Plants supplemented with SNP (100 µM) showed a significant improvement in the growth- and yield-contributing parameters, especially in plants grown under moderate salinity (50 mM NaCl). Essentially, the application of 100 µM SNP remained effective to rescue lentil plants under moderate salinity by regulating plant growth and biochemical pathways. Thus, the exogenous application of SNP could be developed as a useful strategy for improving the performance of lentil plants in salinity-prone environments.

## 1. Introduction

The emerging scenario of climate change and increasing soil salinity (SS) have become serious bottlenecks in attaining high yields of cereals and legume crops [1]. Among abiotic stresses, SS is gradually increasing and establishing rapidly to adversely affect the arable lands and ultimately slicing the available agricultural land and production of crops [1,2]. Salt-stressed plants usually confront toxic ions accumulation and disrupted plant water relations [3]. In addition, SS causes oxidative impairment in crop plants owing to the overabundance of reactive oxygen species (ROS), which are the toxic byproducts of aerobic metabolism [4]. Furthermore, SS suppresses plant development and growth by unbalancing the osmotic adjustments and disturbing hormonal interactions [5]. It disrupts the metabolism in plant cells by inducing water inadequacy, oxygen insufficiency, and nutrients unbalance [6,7]. Under salinity, plants become less dynamic to draw nutrients from the soil and, resultantly, exhibit nutrients deficiency symptoms [8]. If this situation remains unaddressed in the long run, a reduced photosynthetic rate, chlorosis, and necrosis [9] eventually reduce the yield traits, grain yield, and quality [10,11,12]. In such circumstances, the plant life cycle depends upon its antioxidant ability to detoxify the ROS. The plants respond to salinity by stimulating their antioxidant self-defense mechanisms [13], which defend them against abiotic stresses [14]. These defense mechanisms include enzymes like ascorbate peroxidase (APX), catalase (CAT), superoxide dismutase (SOD), and peroxidase (POD), which are biosynthesized in plants, specifically in the chloroplast, and mitochondria [15,16].

Legumes belong to the family of Fabaceae, are an important source of proteins, vegetable oil, vitamins, and minerals [17], and can meet the future food demand in a prevailing climate change scenario. Legumes are very sensitive to salinity stress, which disrupts the nutritional balance, hormonal interactions, and osmotic adjustments that, eventually, affect the overall plant progression from seedling growth to maturity [18]. Among legumes, lentils (*Lens culinaris* Medik.) occupy a pivotal position and serve as an inexpensive source of protein. Similar to other legume crops, lentils are also much vulnerable to salt stress, which may reduce its yield up to 50% [19]. Under increasing soil salinity, it is of the utmost pertinence to find biologically viable ways, which ameliorate the adverse effects of SS and assist lentils to attain optimal vegetative growth along with robust reproductive growth for producing grain yield as per the varietal potential.

Numerous natural and synthetic compounds tend to regulate the biochemical attributes of crop plants under various environmental stresses. Under SS, sodium nitroprusside (SNP), which donates nitric oxide (NO), was found to be effective in promoting wound healing and reducing disease severity in muskmelons [20], inducing resistance against the fungal plant pathogen *Glomerella cingulate* in tea plants [21], against *Monilinia fructicola* in peach [22], against *Botrytis cinerea* in tomato [23], and against the rice black-streaked dwarf virus (RBSDV) [24]. Similarly, the application of SNP enables plants to mitigate abiotic stresses such as moisture stress [25], salinity [26], heat stress [27], chilling stress [28], and heavy metal stress [29]. More explicitly, the use of SNP, under salinity stress, improved the performance of many crop plants like soybean [30], lettuce [31], wheat [4,32], chickpea [33], maize [34], tomato [35], rice [36], and cotton [37]. It can moderate biological, physiological, and chemical reactions occurring in plants under unfavorable environmental conditions [38].

The supplemented SNP holds the potential to mitigate salt stress through the regularization of physiological (chlorophyll content, relative water content, electrolyte leakage, and stomatal behavior) [39] and biochemical (MDA, proline, phenolics, and antioxidant enzymes) attributes of plants [4]. Its role in coping with salinity stress has been well documented; however, its application in lentil crops for salt stress tolerance has not been studied previously. Thus, our research hypothesis was that lentils might respond differently to varying concentrations of SNP, and its dose optimization might alleviate SS effects. Hence, to fill the research gap, the current work was carried out to examine the adverse effects of SS on the growth, yield attributes, and biochemical pathways of lentils along, with assessing the shielding effect of an exogenous application of SNP in different doses.

## 2. Materials and Methods

### 2.1. Experimental Arrangements

The experiment was accomplished in a wire-house during the winter season of 2017–2018. Lentil (cv. Masoor 2009) seeds were obtained from the Cereals Section of the National Agricultural Research Centre (NARC), Islamabad, Pakistan. Earthen pots, having 27 cm height and 24 cm diameter with a filling capacity of 7 kg, were used in this experiment. Peatmoss, pure sand, and sandy loam soil were used in a proportion of 1:3:3 to fill the pots. Prior to sowing, the lentil seeds were surface-sterilized with sodium hypochlorite. Afterwards, ten seeds were initially sown in each pot, and after successful germination, thinned to five seedlings per pot.

### 2.2. NaCl and SNP Treatments

Five-day-old seedlings were exposed to different levels of NaCl (0 mM, 50 mM, and 100 mM) and different concentrations of SNP (0 µM, 50 µM, and 100 µM). The experiment was planned according to a completely randomized design (CRD), having four repeats for each treatment. Both the salinity and SNP treatments were applied weekly from 25 to 70 days after sowing (DAS) to the pots after adding a half-strength Hoagland’s nutrient solution [33]. After 45 days of treatment application, the leaf samples were obtained from all the treatments at 70 DAS and evaluated for biochemical activities of enzymes.

### 2.3. Determination of Growth and Yield Parameters

Out of five seedlings, two seedlings from each pot (at 70DAS) were picked for assessing the biochemical dynamics, and the remaining three seedlings were retained up to maturity for determination of growth, yield-contributing parameters, and grain yield.

The root and shoot lengths of lentil plants were measured with a chrome glass measuring ruler, and their averages were used for further analysis. Similarly, the number of seeds, pods, and branches plant^−1^ were counted for the sampled plants, and the averages were recorded. Plant biomass and seed yield were also determined from three plants; subsequently, their average weight plant^−1^ was used for statistical analysis.

### 2.4. Estimation of Leaf Relative Water Content

The leaf relative water content (*LRWC*) was assessed according to the methodology proposed by [40]. Leaves were weighed after sampling and then immersed in distilled water at room temperature for four hours. After that, leaves were blotted dry and then oven-dried at 80 °C for twenty-four hours. The following formula was used for *LRWC* determination.
$$LRWC\;\left(\%\right)=\frac{FW-DW}{TW-DW}\times 100$$
where *FW* is the fresh weight of the leaves, *DW* is the dry weight of the leaves, and *TW* is the turgid weight of the leaves.

### 2.5. Determination of Leaf Greenness Index

Leaf greenness index was observed by using a portable chlorophyll meter (SPAD-502). For this purpose, five leaves per plant were taken, and their average was used for further analysis.

### 2.6. Estimation of H_2_O_2_ and MDA Content

For measuring H_2_O_2_ content, 500 mg fresh leaf samples were homogenized in trichloroacetic acid (5 mL, 0.1%) and centrifuged for 15 min at 12,000× *g*. After that, supernatant (0.5 mL) was collected, then mixed with a potassium phosphate buffer (0.5 mL of 10 mM having pH 7.0) and potassium iodide (1 mL of 1 M). Finally, the absorbance was measured at 390 nm [41]. For determination of MDA content, fresh leaf samples weighing 500 mg were homogenized with trichloroacetic acid (2.5 mL, 0.1%) and centrifuged for 10 min at 10,000× *g*.

The aliquot was added with trichloroacetic acid (4 mL, 20%) along with thiobarbituric acid (TBA, 0.5%). It was heated for 30 min at 95 °C, then cooled down, and again centrifuged for 15 min at 10,000× *g*. The absorbance was estimated at 532 nm, and the adjustments were taken for undefined turbidity by deducting the absorbance at 600 nm [42].

### 2.7. Antioxidant Enzyme Assays

Fresh leaf samples (500 mg) were mashed in a chilled phosphate buffer (0.1 M, pH 7.5) and EDTA (0.5 mM), followed by centrifuge for 10 min at 12,000× *g*. After that, the obtained supernatant was utilized for the estimation of antioxidant enzymes.

The activity of SOD was estimated by quantifying the ability of the enzyme, which inhibits the photochemical reduction of nitro-blue tetrazolium chloride [43]. The absorbance was read at 560 nm, whereas one unit of SOD activity (EU) was defined as the quantity of enzyme needed for 50% inhibition of the nitroblue tetrazolium (NBT) photoreduction rate. Catalase activity was measured by using the method of [44], whereas the change in absorbance was read out at 240 nm. The activity of peroxidase was estimated according to [45]. The absorbance was determined at 420 nm.

### 2.8. Statistical Analysis

The data were analyzed using the computer software Statistix 8.1 (Analytical Software, Tallahassee, FL, USA), and means were compared by employing the least significant difference (LSD) test at *p* ≤ 0.05. Graphic outputs were made in Microsoft Office 2016 by using MS Excel.

## 3. Results

Our results exhibited that imposed salinity caused considerable damage to the lentil plants by reducing their growth traits like shoot and root length, relative water content, and chlorophyll content of leaf (Table 1). The adverse effects of salinity were increased with an increasing level of salinity as in the severe salinity-stressed treatment (NaCl-100 mM + SNP-0 µM), which was followed by moderate salinity-stressed treatment (NaCl-50 mM + SNP-0 µM). The severe salinity stress imparted a significant influence on shoot and root length, which were decreased by 41 and 63%, respectively. When comparing T_2_ (moderate salinity treatment) with T_4_ and T_6_ (moderate salinity treated with different levels of SNP), it was observed that all the parameters showed gradual improvement as the concentration of SNP was raised from 50 µM to 100 µM. Similarly, when comparing T_3_ (severe salinity) with T_5_ and T_7_ (severe salinity treated with different levels of SNP), it was found that the performance of lentil plants in terms of studied parameters was enhanced by increasing the level of SNP from 50 µM to 100 µM. In comparison, the relative water content and leaf greenness index were decreased by 47% and 42%, respectively (Table 1).

The salinity alleviating effects of SNP were evident in the treatments where moderate salinity stress (50 mM NaCl) was treated with the exogenous application of SNP (Table 1). Considerable improvement was observed in treatment the T_6_, where all the growth parameters were improved significantly by applying 100 µM SNP, as compared to the severe salinity-stressed treatment (T_3_), where all studied parameters like the shoot length, root length, leaf relative water content, and leaf greenness index were increased by 65, 142, 97, and 103%, respectively, as compared to T_3_ treatment (Table 1).

Significant reduction in yield-contributing parameters took place with the increase in the intensity of salinity (Table 2). As compared to the control treatment (T_1_), the decline was at the maximum in the severe salinity-stressed treatment (T_3_), where the number of branches, pods, and seed per plant was reduced by 53, 45, and 60%, respectively, while the plant seed yield and biomass were reduced by 60 and 54%, respectively. Plants supplemented with different levels of SNP showed significant tolerance to the induced salinity levels. Plants subjected to the highest dose of SNP possessed the maximum tolerance against the moderate level of salinity (50 mM). In this treatment, plant seed yield and biomass were increased by 211 and 235%, respectively, as compared to the severe salinity-stressed treatment. Similarly, the application of SNP increased the number of branches, pods, and seed per plant, compared to the severe salinity-stressed treatment by 196, 144, and 208%, respectively (Table 2).

The biochemical performances of lentils in terms of MDA, H_2_O_2_, SOD, CAT, and POD biosynthesis under different levels of salinity and SNP are presented in Figure 1 and Figure 2. Both the figures revealed that the concentration of these compounds was higher in plants exposed to severe salinity-stressed treatment. However, the highest increment of MDA content (68%) over the control treatment (T_1_) was found for 100 Mm salinity levels, where SNP was not applied as a foliage spray.

Similarly, in comparison with the control treatment, the H_2_O_2_ content was increased by 62 and 137% in plants exposed to 50 and 100 mM salinity levels, respectively, with no foliar application of SNP. The increment of SNP in the salinity-stressed treatments gradually reduced the building up of MDA and H_2_O_2_. The treatment, involving 50 mM salinity level and foliage-applied SNP (100 µM), showed the maximum reduction of MDA (49%) and H_2_O_2_ (65%), as compared to the severe salinity-stressed treatment (Figure 1). Correspondingly, a rapid increase in the activities of enzymes SOD, CAT, and POD was noted in treatments involving 50 and 100 mM of salinity levels with the absence of SNP application. When compared with the control treatment, the SOD, CAT, and POD were increased by 88, 84, and 86%, respectively, by the 100 mM salinity treatment. Contrariwise, the amount of SOD, CAT, and POD were reduced by 33, 29, and 34%, respectively, in T_6_, which was supplemented with a higher amount of SNP (Figure 2).

## 4. Discussion

In most crop plants, the seedling phase is the most vulnerable to harsh ecological conditions. The legume crops, especially lentils, is very sensitive to saline conditions at seedling growth stage [46]. The results of our research trial were in accordance with the postulated hypothesis as salinity stress severely hampered the growth, biochemical processes as well as yield attributes of lentil plants by disrupting the normal functioning and performance of plants [47]. The SS negatively affected the water retention and turgidity of the plant cells [48], modulation of stomata, photosynthetic capacity, and ultimately declined in the growth and yield of plants became evident [49].

Under salinity stress, vegetative and reproductive growth of plants got adversely affected and revamping of biochemical pathways through exogenous application of salinity alleviating chemicals might enable plants to cope with adverse effects of abiotic stresses. In this study, plants under different intensities of NaCl stress produced significantly shorter roots and shoots, which was in line with the results of [4,32] in wheat [50] and rice. The addition of SNP to the plants under different intensities of salt stress markedly amended their shoot and root length (Table 1). Similar results were also obtained by [51,52] while working on wheat and safflower, respectively. It was found that nitric oxide reacted with phospholipids and modified the cell wall, which ultimately recovered the plant growth in stressful environments [53].

The photosynthetic capacity depends upon the amount of chlorophyll present in the plant leaves, while SS significantly degrades chlorophyll [54]. Similarly, our results highlighted a significant reduction in the greenness index of lentil leaves under severe salinity-stressed treatment (100 Mm NaCl), which led to a noticeable decline in the seed yield. The relative water content of the leaf was lowest in severe salinity-stressed treatment (Table 1), and it might be attributed to the disrupted plant root system by choking water absorption under higher NaCl concentration [55]. In addition to this, the remedial effects of nitric oxide were vibrant in the treatments where SNP was supplemented with NaCl, and lentil plants performed better to improve their leaf relative water and greenness index. These results also confirmed the early findings of [55] and [56] in mustard and rice, respectively, who observed that the water content and chlorophyll content of leaves were increased with the supplementation of nitric oxide under a saline environment. The application of nitric oxide was found effective in enhancing the formation of chlorophyll pigments, which were vital for triggering the photosynthesis rate through shielding the membranes of the cell organelles comprising chlorophyll [57].

Yield losses in different legume crops were recorded from 12 to 100% [58]. The reduction in yield-contributing parameters were significant under higher NaCl concentrations. The effects of salinity were more pronounced for 100 Mm NaCl stress, which caused a significant decline in the yield attributes, while this treatment was followed by a moderate level of salinity (Table 2). These findings were in agreement with [59], who also reported a noticeable drop in yield-related attributes in soybeans exposed to a severe salt-stress environment. Salinity caused a 20% reduction in the seed yield of chickpeas due to wrinkled seeds [60] and lower seed weight due to lesser grains per pod in mung bean [61].

Salinity stress interrupts metabolic pathways, including several enzymes and antioxidants, by accretion of unwanted reactive oxygen species, which damage plant cells and ultimately plant growth and development [13]. The current study revealed that MDA and H_2_O_2_ concentrations were increased in severe a salinity-stressed treatment compared to control treatment. Contrarily, these parameters showed the minimum corresponding values, where the highest dose of SNP (100 mM) was added to cope with moderate salinity (Figure 1). Similar results were also obtained in chickpeas [33] and wheat [62]. It might be inferred that the SNP application as foliar spray might potentially impart tolerance against salinity by enabling the plants to make metabolic adjustments for reducing the levels of MDA and H_2_O_2_ [63]. Optimizing the production of MDA and H_2_O_2_ to normal limits (as in T_1_ Figure 1) endorsed the shielding role of SNP to confrontation salinity.

Plants also have a high tendency to activate their defensive system by overproducing proline, APX, CAT, SOD, and GR as their self-defense mechanism under environmental severities [33]. The amount of SOD, CAT, and POD was found to be increased as the intensity of salinity was increased from 0 to 100 mM (Figure 2), while their corresponding values were declined significantly with an increasing concentration of SNP, which was applied as a foliar spray (Figure 2). Moreover, it was noticed that the SNP application (100 µM) on plants exposed to 50 mM salinity level exhibited the lowest values for SOD, CAT, and POD, and these values were closer to the normal ranges as observed in the control treatment. These results proved the ameliorating effects of SNP under different levels of salinity stress in lentils, which supported the earlier studies in different crops, as reported by [64] in mustard, [65] in cotton, and [35] in tomato.

## 5. Conclusions

The research trial was executed to determine the effects of varying levels of salinity on lentil growth, yield attributes, and biochemical functioning, along with exploiting the role of SNP in alleviating the harmful effects of salinity. The research findings were in line with the postulated hypothesis as salinity reduced the physiological functions, growth, yield traits, and yield of lentils to a great extent. The highest dose of SNP (100 µM) remained effective in accelerating and modifying plant growth and yield by optimizing their biochemical and metabolic functioning in a salinity-prone environment. Therefore, it was recommended to use exogenous applications of SNP to obtain a better yield of lentils in moderate saline areas. However, it was also suggested to evaluate other concentrations of SNP for mitigating the harmful influence of salinity on lentils.

## Figures and Tables

**Figure 1 molecules-26-02576-f001:**
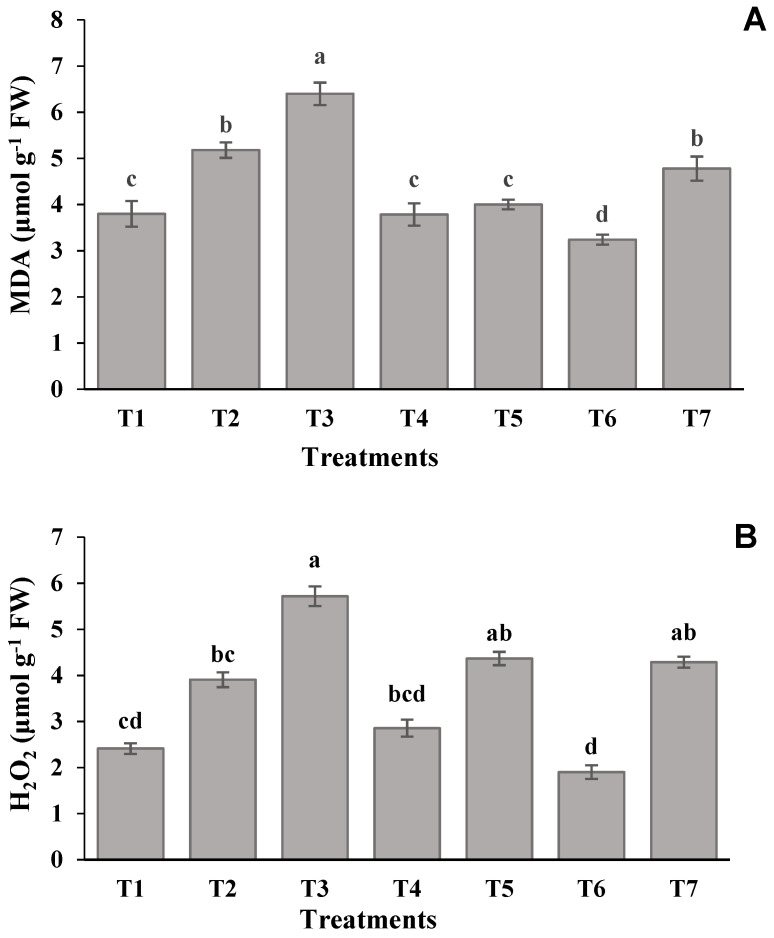
Effect of SNP on the (**A**) MDA content and (**B**) H_2_O_2_ content in leaves of lentil plants under salinity stress. Data presented are the means ± SE (*n* = 4). Different letters on the bars indicate significant differences among the treatments at *p* ≥ 0.05 for MDA and H_2_O_2_ content. T1 = 0 mM of NaCl + 0 µM of SNP (control); T2 = 50 mM of NaCl+ 0 µM of SNP; T3 = 100 mM of NaCl + 0 µM of SNP; T4 = 50 mM of NaCl + 50 µM of SNP; T5 = 100 mM of NaCl +50 µM of SNP; T6 = 50 mM of NaCl + 100 µM of SNP; T7 = 100 mM of NaCl+ 100 µM of SNP.

**Figure 2 molecules-26-02576-f002:**
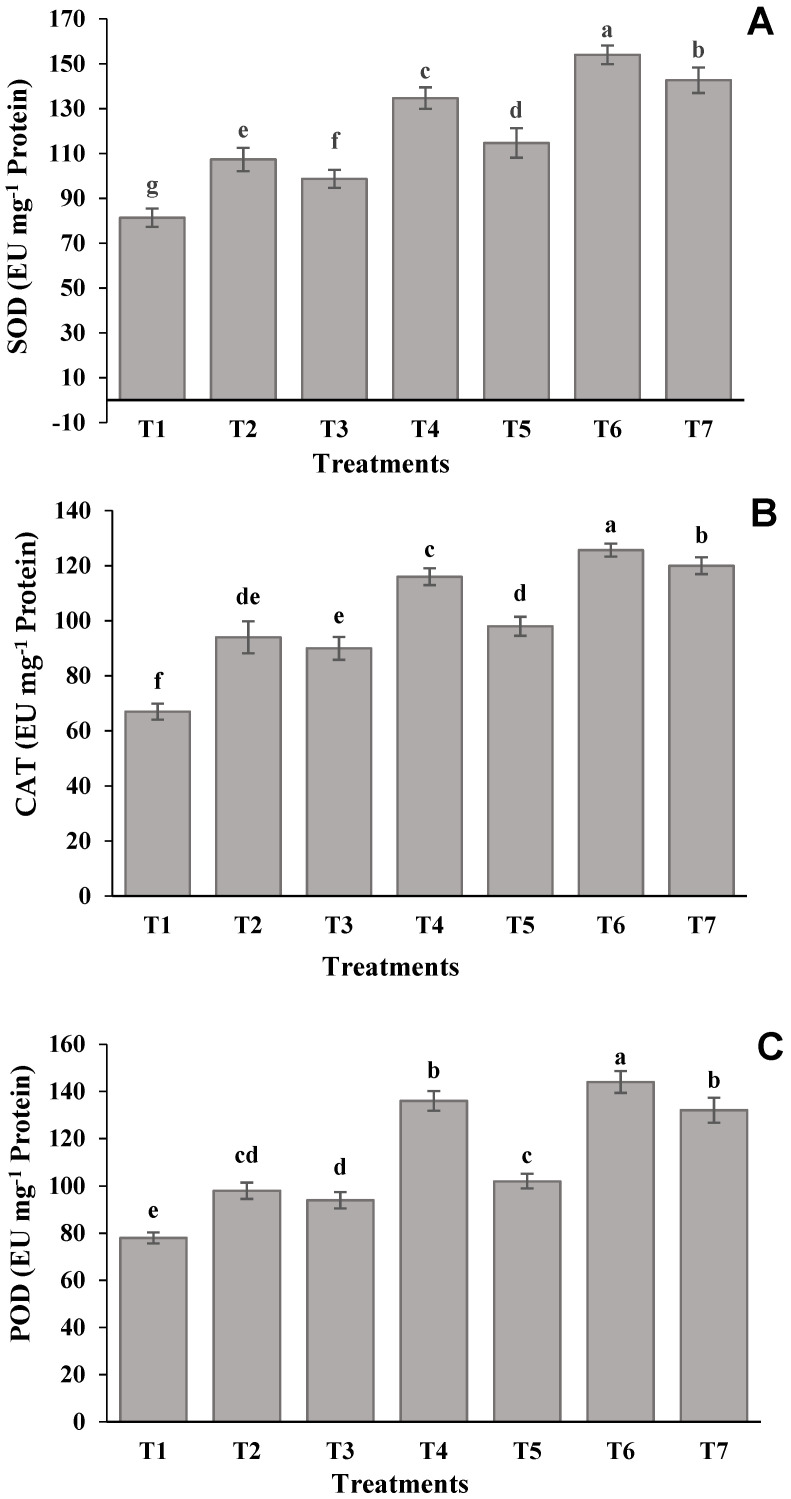
Effect of SNP on the antioxidant enzymes activity (**A**) SOD, (**B**) CAT, and (**C**) POD in leaves of lentil plants under salinity stress. Data presented are the means ± SE (*n* = 4). Different letters on the bars indicate significant differences among the treatments at *p* ≥ 0.05 for individual antioxidant enzymes. T1 = 0 mM of NaCl + 0 µM of SNP (control); T2 = 50 mM of NaCl + 0 µM of SNP; T3 = 100 mM of NaCl + 0 µM of SNP; T4 = 50 mM of NaCl + 50 µM of SNP; T5 = 100 mM of NaCl + 50 µM of SNP; T6 = 50 mM of NaCl + 100 µM of SNP; T7 = 100 mM of NaCl + 100 µM of SNP.

**Table 1 molecules-26-02576-t001:** Effect of SNP on shoot, root, and leaf attributes of lentils grown under different NaCl levels.

Treatments	Shoot Length (cm)	Root Length (cm)	Leaf Relative Water Content (%)	Leaf Greenness Index
T_1_ = 0 mM of NaCl + 0 µM of SNP (Control)	40.12 ± 1.18 a	20.82 ± 0.62 a	81.58 ± 1.88 a	12.51 ± 0.97 ab
T_2_ = 50 mM of NaCl + 0 µM of SNP	32.18 ± 0.83 c	12.55 ± 0.95 de	64.48 ± 2.05 d	9.42 ± 1.22 cd
T_3_ = 100 mM of NaCl + 0 µM of SNP	23.41 ± 1.24 e	7.62 ± 0.76 f	42.86 ± 1.06 e	7.28 ± 1.05 d
T_4_ = 50 mM of NaCl + 50 µM of SNP	36.66 ± 1.14 b	16.23 ± 0.96 bc	74.27 ± 1.71 b	12.27 ± 1.07 ab
T_5_ = 100 mM of NaCl + 50 µM of SNP	30.55 ± 0.85 c	13.71 ± 0.83 cd	69.11 ± 1.68 c	10.24 ± 0.87 bc
T_6_ = 50 mM of NaCl + 100 µM of SNP	38.72 ± 1.02 ab	18.42 ± 0.73 ab	84.32 ± 2.46 a	14.80 ± 0.97 a
T_7_ = 100 mM of NaCl + 100 µM of SNP	27.18 ± 0.82 d	10.22 ± 0.87 ef	65.33 ± 1.99 d	11.71 ± 0.99 bc

The data presented are the mean values ± standard deviation (*n* = 4). Different letters next to the mean value indicate significant differences at *p* ≤ 0.05.

**Table 2 molecules-26-02576-t002:** Effect of SNP on the yield attributes of lentils grown under different levels of NaCl.

Treatments	Number of Branches Plant^−1^	Number of Pods Plant^−1^	Number of Seeds Plant^−1^	Seed Yield (g Plant^−1^)	Biomass (g Plant^−1^)
T_1_ = 0 mM of NaCl + 0 µM of SNP	10.50 ± 0.53 b	36.52 ± 2.50 b	54.50 ± 3.18 b	1.32 ± 0.18 ab	9.17 ± 0.73 b
T_2_ = 50 mM of NaCl + 0 µM of SNP	7.25 ± 0.80 c	27.51 ± 2.11 d	34.52 ± 1.97 e	0.86 ± 0.04 ab	7.82 ± 0.58 b
T_3_ = 100 mM of NaCl + 0 µM of SNP	4.91 ± 0.69 d	20.04 ± 1.73 e	21.54 ± 1.06 f	0.53 ± 0.06 b	4.22 ± 0.71 c
T_4_ = 50 mM of NaCl + 50 µM of SNP	8.66 ± 0.82 c	34.54 ± 1.52 bc	49.48 ± 2.71 c	1.23 ± 0.08 ab	8.95 ± 0.84 b
T_5_ = 100 mM of NaCl + 50 µM of SNP	7.33 ± 0.35 c	32.20 ± 2.09 c	39.50 ± 2.44 d	0.93 ± 0.12 ab	5.42 ± 0.62 c
T_6_ = 50 mM of NaCl + 100 µM of SNP	14.58 ± 0.72 a	48.75 ± 2.81 a	66.25 ± 2.70 a	1.65 ± 0.21 a	14.15 ± 1.06 a
T_7_ = 100 mM of NaCl + 100 µM of SNP	8.16 ± 1.01 c	33.75 ± 2.13 bc	48.52 ± 2.32 c	1.21 ± 0.13 ab	8.38 ± 0.74 b

Data presented are the mean values ± standard deviation (*n* = 4). Different letters next to the mean value indicate significant differences at *p* ≤ 0.05.

## Data Availability

Not applicable.

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
