# Peer review of "Exogenous Sodium Nitroprusside Mitigates Salt Stress in Lentil (Lens culinaris Medik.) by Affecting the Growth, Yield, and Biochemical Properties"

_molecules, 2021, doi:10.3390/molecules26092576_

Round 1

Reviewer 1 Report

The work studies the use of sodium nitroprusside to mitigate the effects of salinity stress. For that purpose, plant growth and yield-related attributes, biochemistry of enzymes, such as superoxide dismutase, catalase, and peroxidase, and amassing of leaf malondialdehyde and hydrogen peroxide have been studied. The results show that the effects of moderate salinity stress can be counteracted using sodium nitroprusside

Firstly, the introduction shows a wide background of the topic and a properly number of references has been used. However, in my opinion, some minor changes could improve it.

  • Line 84: Please indicate some examples of abiotic and biotic stress where an improvement by SNP has been described
  • Line 89: What physiological and biochemical pathways of plants are regularized by SNP? Please provide some references.
  • Line 91: Has the application of SNP been previously studied in lentil crops for tolerance to salt stress? If yes. please provide some reference.

The experiments are well design and carried out properly to demonstrate that SNP could be used to obtain better yield of lentil in moderate saline areas. However, in several parts of the result section the effects of SNP over moderate and severe salinity are mixed and compared. For example, in lines 173-179 or 187-190 the effect of SNP in moderate salinity (T6) is compared with the severe salinity control (T3). The improvement of the parameters should only be compared at the same salt concentrations. (T2 with T4 and T6 and T3 with T5 and T7). It could also be interesting to compare the parameters after the treatments and the T1 control, measuring the percentage of the parameters recovered by the SNP.

Minor points:

  • Line 127: How did the plants dry?
  • Line 141: Please indicate the chemical name of the reagent TBA

Author Response

Response to Reviewer’s Comments (Reviewer 1)

Dear Sir,

Thanks very much for your efforts and useful comments about our manuscript. We are very grateful and appreciate your good comments that help us to make the paper more quality and accurate. We responded to all the comments of the reviewers and heighted the changes in the manuscript, so anyone can follow the corrections. We have modified the manuscript accordingly, and the detailed corrections are listed below point by point and we hope this edition is more suitable and cover all the comments.

Comments:

               The work studies the use of sodium nitroprusside to mitigate the effects of salinity stress. For that               purpose, plant growth and yield-related attributes, biochemistry of enzymes, such as superoxide          dismutase, catalase, and peroxidase, and amassing of leaf malondialdehyde and hydrogen                peroxide have been studied. The results show that the effects of moderate salinity stress can be counteracted using sodium nitroprusside

               Firstly, the introduction shows a wide background of the topic and a properly number of references           has been used. However, in my opinion, some minor changes could improve it.

Response:

               We are thankful to the reviewer for giving ample time to review the manuscript. The suggestions        and deficiencies pointed out by the reviewer are positively rectified in the updated version.

Comment:         

Line 84: Please indicate some examples of abiotic and biotic stress where an improvement                               by SNP has been described.

Response:

We have added the examples of use of SNP to improve biotic stresses (disease and pest) and        abiotic stresses (drought, salinity, heat stress, chilling stress and heavy metal stress) along with       recent citations in the suggested liens.

Comment:

Line 89: What physiological and biochemical pathways of plants are regularized by SNP? Please provide some references.

Response:

               We have revised this part by adding the suggested examples and references regarding                physiological (chlorophyll content, relative water content, electrolyte leakage and stomatal behavior)          and biochemical (MDA, proline, phenolics and antioxidant enzymes) attributes of plants.

Comment:

               Line 91: Has the application of SNP been previously studied in lentil crops for tolerance to salt    stress? If yes. please provide some reference.

Response:

The application of SNP in lentil for salinity tolerance has never been studied previously. Hence, I have found one study wherein SNP was used to study the effects of nitric oxide (NO) on the germination of lentil seeds and early seedling growth. This is from a thesis.

URL: http://acikerisim.harran.edu.tr:8080/jspui/handle/11513/852

Hence, we also amend the sentence as “, its application in lentil crop for salt stress tolerance has not been studied previously”.

Comments:

The experiments are well design and carried out properly to demonstrate that SNP could be used to obtain better yield of lentil in moderate saline areas. However, in several parts of the result section the effects of SNP over moderate and severe salinity are mixed and compared. For example, in lines 173-179 or 187-190 the effect of SNP in moderate salinity (T6) is compared with the severe salinity control (T3). The improvement of the parameters should only be compared at the same salt concentrations. (T2 with T4 and T6 and T3 with T5 and T7). It could also be interesting to compare the parameters after the treatments and the T1 control, measuring the percentage of the parameters recovered by the SNP.

Response:

               We have revised the result section. The revisions have been made as suggested.

Minor points:

Comment:

Line 127: How did the plants dry?

Response:

               Leaves were weighed after sampling, and then immersed in distilled water at room temperature for                four hours. After that, leaves were blotted dried and then oven dried at 80 °C for twenty four hours.

               This information is also added in the materials and methods for determination of relative water    content.

Comment:

Line 141: Please indicate the chemical name of the reagent TBA

Response:

It is Thiobarbituric acid (TBA). This information has now been added in the text.

Reviewer 2 Report

To authors:

- Major Compulsory Revisions:

1) The results in this manuscript (Ms) do not match the criteria for publication. This paper lacks novelty and originality in terms of new compounds, new methods, new pathways, etc. The presented data are preliminary, because role of sodium nitroprusside (SNP) in coping with salinity stress, has been well documented in many crops. The presented results are like a short communication, not an Original Article.

The authors should add new experiments aimed at studying the mechanism of action of SNP, like expression of the regulatory genes or etc.

2) Explain why the concentration “100 mM of NaCl+100 μM of SNP” decreased shoot length and root length (Table 1) compared with concentration “100 mM of NaCl+50 μM of SNP”, but at the same time “100 mM of NaCl+100 μM of SNP” increased biomass (Table 2) compared with concentration “100 mM of NaCl+50 μM of SNP”.

3) Also, I recommend that the authors should use some help of a native English speaker or send the Ms to an English Editing Service that proofreads scientific writing.

Minor:

4) Line 38: “@ 50 mM”, I did not understand “@”, the authors should remove “@” in the Ms text.

Author Response

Response to Reviewer’s Comments (Reviewer 2)

Dear Sir,

Thanks very much for your efforts and useful comments about our manuscript. We are very grateful and appreciate your good comments that help us to make the paper more quality and accurate. We responded to all the comments of the reviewers and heighted the changes in the manuscript, so anyone can follow the corrections. We have modified the manuscript accordingly, and the detailed corrections are listed below point by point and we hope this edition is more suitable and cover all the comments.

- Major Compulsory Revisions:

Comment:

The results in this manuscript (Ms) do not match the criteria for publication. This paper lacks novelty and originality in terms of new compounds, new methods, new pathways, etc. The presented data are preliminary, because role of sodium nitroprusside (SNP) in coping with salinity stress, has been well documented in many crops. The presented results are like a short communication, not an Original Article.

Response:

               No doubt a lot of work has been done on many crops to alleviate salinity and other abiotic stress through SNP. Whereas, there is no evidence of using SNP on Lentil crop to mitigate salinity stress.    As Lentil (Lens culinaris L) is an important legume crop, and serves as an inexpensive source of protein to human food. In South Asia, especially in Pakistan, it is considered as poor man’s meat. While during cultivation, it is given less importance and planted on marginal lands with poor fertility status. Likewise, other legume crops, lentil is also much vulnerable to salt stress, which reduces its yield up to 50%. Therefore we focused on this crop to improve its yield while cultivating on lands with moderate salinity issues.

               We think that it is an important study to improve the yield of lentil with the application of an authentic salinity mitigating agent i.e. SNP.

Comment:

The authors should add new experiments aimed at studying the mechanism of action of SNP, like expression of the regulatory genes or etc.

Response:

               It is a nice suggestion to study the expression of the regulatory genes linked with SNP. We may plan for this study in coming crop season.  

Comment:

Explain why the concentration “100 mM of NaCl+100 μM of SNP” decreased shoot length and root length (Table 1) compared with concentration “100 mM of NaCl+50 μM of SNP”, but at the same time “100 mM of NaCl+100 μM of SNP” increased biomass (Table 2) compared with concentration “100 mM of NaCl+50 μM of SNP”.

Response:

               You are right as the results revealing the same as pointed by you. We think, considering shoot and root length, it is not a big difference as it should be seen with number of branches and other growth attributes.

While on the other hand, biomass improved by adding more SNP under severe salinity conditions as compared to the treatment where less SNP was added to the severe salinity conditions. These results are straight forward as no. of branches, number of pods and number of seeds collectively contribute to the biomass.

Comment:

Also, I recommend that the authors should use some help of a native English speaker or send the Ms to an English Editing Service that proofreads scientific writing.

Response:

One of the co-author is a native English speaker and he has completely revised the whole manuscript. Now the English language has been improved.

Minor:

Comment:

 Line 38: “@ 50 mM”, I did not understand “@”, the authors should remove “@” in the Ms text.

Response:

               We have used @ which means at the rate of. We used it to give a new look and to avoid similarity.

Reviewer 3 Report

The results are interesting. However, the manuscript was not written in the format of a scientific paper, and there are many findings but few in-depth discussions or applications. The manuscript could be resubmitted after a revision. 

The authors did not prepare a solution without NaCl  and with  sodium nitroprusside (SNP)  as a control.
To analyze the degradation process of chlorophyll by NaCl, different NaCl concentrations were used to monitor the degradation of chlorophyll. Considering the chlorophyll is very sensitive to heat and light as well, the experimental conditions to protect chlorophyll from other possible ways of degradation should be given. Also, it's better to set the relative control groups for comparison.
I don't understand leaf  chlorophyll (%). The chlorophyllometer measures the leaf greenness index, this is not the chlorophyll content. There is no method of determining protein content in the manuscript.
Have you done one experiment only?

Author Response

Response to Reviewer’s Comments (Reviewer 3)

The results are interesting. However, the manuscript was not written in the format of a scientific paper, and there are many findings but few in-depth discussions or applications. The manuscript could be resubmitted after a revision. 

Dear Sir,

Thanks very much for your efforts and useful comments about our manuscript. We are very grateful and appreciate your good comments that help us to make the paper more quality and accurate. We responded to all the comments of the reviewers and heighted the changes in the manuscript, so anyone can follow the corrections. We have modified the manuscript accordingly, and the detailed corrections are listed below point by point and we hope this edition is more suitable and cover all the comments.

Comment:

The authors did not prepare a solution without NaCl and with  sodium nitroprusside (SNP)  as a control.

Response:

               It is to clarify that we used first treatment where in no NaCl and no SNP was used and it is considered as control treatment. Furthermore we use SNP in combination with moderate to high NaCl. Our focus was to study the function of SNP to promote plant growth and physiological functioning under varying intensities of salinity. Therefore we made different combination of NaCl+SNP and compared them with the control.

Comment:

To analyze the degradation process of chlorophyll by NaCl, different NaCl concentrations were used to monitor the degradation of chlorophyll. Considering the chlorophyll is very sensitive to heat and light as well, the experimental conditions to protect chlorophyll from other possible ways of degradation should be given.

Response:

               You are right that chlorophyll is very sensitive to heat and light. In this study we mainly focused on salinity stress, therefore other conditions of light and heat were uniform for the entire set of treatments.

Comment:

Also, it's better to set the relative control groups for comparison.

Response:

               We have revised the results and discussion section to give better presentation by comparing the treatments with the control as well as the application of moderate concentration of SNP with moderate salinity stress and high concentration of SNP with severe salinity.

Comment:

I don't understand leaf chlorophyll (%). The chlorophyllometer measures the leaf greenness index, this is not the chlorophyll content.

Response:

               It is a typographic mistake. Thank you for this correction. We used SPAD-502 chlorophyll meter. We have replaced chlorophyll content with leaf greenness index in the entire manuscript.

Comment

There is no method of determining protein content in the manuscript.

Response:

               We could not determine protein content so it is not given.

Comment:

Have you done one experiment only?

Response:

               Yes we have done one experiment under control conditions. In which we have studied growth, physiological and biochemical attributes of lentil crop under different intensities of salinity and the use of different rates of SNP to alleviate plant growth under this abiotic stress.

Round 2

Reviewer 1 Report

Authors have introduced several changes in the manuscript considering the comments. However, some spaces are missing in some words, for example compoundstend in line 82, onshoot and lengthwhich in lines 179 and 180, etc. Please, this text editing should be made before published.

Author Response

Dear Sir,

Thanks very much for your efforts and useful comments about our manuscript. We are very grateful and appreciate your good comments that help us to make the paper more quality and accurate. We responded to all the comments of the reviewers and heighted the changes in the manuscript, so anyone can follow the corrections. We have modified the manuscript accordingly, and the detailed corrections are listed below point by point and we hope this edition is more suitable and cover all the comments.

Authors have introduced several changes in the manuscript considering the comments. However, some spaces are missing in some words, for example compoundstend in line 82, onshoot and lengthwhich in lines 179 and 180, etc. Please, this text editing should be made before published.

Response: We are thankful to the reviewer for considering the revised version of this manuscript. We have corrected these typographic mistakes, Moreover we have thoroughly read out the manuscript for such kind of mistakes and have corrected them all.

Line 82: compoundstend…………..compounds tend

Line 179: onshoot……….on shoot

Line 180: lengthwhich………….length which

Reviewer 2 Report

- Major Compulsory Revisions:

1) Firstly, I want to indicate, that Molecules is journal with high Impact Factor, therefore, authors should improve experimental data presented in manuscript (Ms) text and should improve discussion and conclusions. This paper lacks novelty and originality in terms of new compounds, new methods, new pathways, etc. The presented data are preliminary, because role of sodium nitroprusside (SNP) in coping with salinity stress has been well documented in many crops.

The authors should add new experiments aimed at studying the mechanism of action of SNP, like expression of the regulatory genes or etc. I did not find any new data in the revised Ms version. All experimental data in the Ms based on two small Tables and two small Figures.

- Minor:

2) Line 38: “@ 50 mM”, I did not understand “@”, the authors should remove “@” in the Ms text. I advise you to remove “@” from the Ms text.

3) Table 1 and 2 – authors should present standard error of the mean or standard deviation.

Author Response

Dear Sir,

Thanks very much for your efforts and useful comments about our manuscript. We are very grateful and appreciate your good comments that help us to make the paper more quality and accurate. We responded to all the comments of the reviewers and heighted the changes in the manuscript, so anyone can follow the corrections. We have modified the manuscript accordingly, and the detailed corrections are listed below point by point and we hope this edition is more suitable and cover all the comments.

Firstly, I want to indicate, that Molecules is journal with high Impact Factor, therefore, authors should improve experimental data presented in manuscript (Ms) text and should improve discussion and conclusions. This paper lacks novelty and originality in terms of new compounds, new methods, new pathways, etc. The presented data are preliminary, because role of sodium nitroprusside (SNP) in coping with salinity stress has been well documented in many crops.

The authors should add new experiments aimed at studying the mechanism of action of SNP, like expression of the regulatory genes or etc. I did not find any new data in the revised Ms version. All experimental data in the Ms based on two small Tables and two small Figures.

Response:

               No doubt a lot of work has been done on many crops to alleviate salinity and other abiotic stress through SNP. Whereas, there is no evidence of using SNP on Lentil crop to mitigate salinity stress.    As Lentil (Lens culinaris L) is an important legume crop, and serves as an inexpensive source of protein to human food. In South Asia, especially in Pakistan, it is considered as poor man’s meat. While during cultivation, it is given less importance and planted on marginal lands with poor fertility status. Likewise, other legume crops, lentil is also much vulnerable to salt stress, which reduces its yield up to 50%. Therefore we focused on this crop to improve its yield while cultivating on lands with moderate salinity issues.

It is a nice suggestion to study the expression of the regulatory genes linked with SNP. We may plan for this study in coming crop season.  

               We think that it is an important study to improve the yield of lentil with the application of an authentic salinity mitigating agent i.e. SNP.

- Minor:

2) Line 38: “@ 50 mM”, I did not understand “@”, the authors should remove “@” in the Ms text. I advise you to remove “@” from the Ms text.

Response: We have made revisions by replacing “@ 50 mM” with at the concentration of 50 mM and 100 mM. As per your suggestions, we have carefully read and made such changes in the entire manuscript.

3) Table 1 and 2 – authors should present standard error of the mean or standard deviation.

Response: We have added Standard deviation in both tables 1 and 2.

Reviewer 3 Report

Accept in present form

Author Response

Dear Sir,

Thanks very much for your efforts and useful comments about our manuscript. We are very grateful and appreciate your good comments that help us to make the paper more quality and accurate.

Accept in present form

Response: We are thankful to the reviewer for his valuable time and suggestions to improve this manuscript.
